# Down Regulation of *Catsper1* Expression by Calmodulin Inhibitor (Calmidazolium): Possible Implications for Fertility

**DOI:** 10.3390/ijms23158070

**Published:** 2022-07-22

**Authors:** Angela Forero-Forero, Stephany López-Ramírez, Ricardo Felix, Javier Hernández-Sánchez, Emiliano Tesoro-Cruz, Sandra Orozco-Suárez, Janet Murbartián, Elizabeth Soria-Castro, Aleida Olivares, Carolina Bekker-Méndez, Vladimir Paredes-Cervantes, Norma Oviedo

**Affiliations:** 1Centro de Investigación y de Estudios Avanzados del Instituto Politécnico Nacional (Cinvestav-IPN), Departamento de Biología Celular, Ciudad de México 07360, Mexico; angelavfore@ciencias.unam.mx (A.F.-F.); rfelix@cinvestav.mx (R.F.); 2Instituto Mexicano del Seguro Social (IMSS), Hospital General de Zona Núm. 68, Ecatepec 55400, Mexico; fanny300494@hotmail.com; 3Centro de Investigación y de Estudios Avanzados del Instituto Politécnico Nacional (Cinvestav-IPN), Departamento de Genética y Biología Molecular, Ciudad de México 07360, Mexico; javierh@cinvestav.mx; 4Instituto Mexicano del Seguro Social (IMSS), Hospital de Infectología del Centro Médico Nacional La Raza, Unidad de Investigación Médica en Inmunología e Infectología, Ciudad de México 02990, Mexico; emiliano_tesoro@hotmail.com (E.T.-C.); bekkermendez@yahoo.com (C.B.-M.); vlapace@hotmail.com (V.P.-C.); 5Instituto Mexicano del Seguro Social (IMSS), Centro Médico Nacional siglo XXI, Hospital de Especialidades, Unidad de Investigación Médica en Enfermedades Neurológicas, Ciudad de México 06720, Mexico; sorozco5@hotmail.com; 6Centro de Investigación y de Estudios Avanzados del Instituto Politécnico Nacional (Cinvestav-IPN), Sede sur, Departamento de Farmacobiología, Ciudad de México 14330, Mexico; murbartian@cinvestav.mx; 7Instituto Nacional de Cardiología “Ignacio Chavéz”, Departamento de Biomedicina Cardiovascular, Ciudad de México 14080, Mexico; elizabethsoria824@gmail.com; 8Instituto Mexicano del Seguro Social (IMSS), Hospital de Gineco Obstetricia No. 4 Luis Castelazo Ayala, Unidad de Investigación Médica en Medicina Reproductiva, Ciudad de México 01090, Mexico; aleidaolivares@gmail.com

**Keywords:** *Catsper*, calmidazolium, male fertility

## Abstract

The CatSper channel localizes exclusively in the flagella of sperm cells. The Catsper1 protein, together with three pore units, is essential for the CatSper Channel formation, which produces flagellum hyperactivation and confers sperm fertility. *Catsper1* expression is dependent on Sox transcription factors, which can recognize in vitro at least three Sox binding sites on the promoter. Sox transcription factors have calmodulin-binding domains for nuclear importation. Calmodulin (CaM) is affected by the specific inhibitor calmidazolium (CMZ), which prevents the nuclear transport of Sox factors. In this work, we assess the regulation of the *Catsper1* promoter in vivo by Sox factors in the murine testis and evaluate the effects of the inhibitor calmidazolium on the expression of the *Casper* genes, and the motility and fertility of the sperm. *Catsper1* promoter has significant transcriptional activity in vivo; on the contrary, three Sox site mutants in the *Catsper1* promoter reduced transcriptional activity in the testis. CaM inhibition affects Sox factor nuclear transport and has notable implications in the expression and production of *Catsper1*, as well as in the motility and fertility capability of sperm. The molecular mechanism described here might conform to the basis of a male contraceptive strategy acting at the transcriptional level by affecting the production of the CatSper channel, a fundamental piece of male fertility.

## 1. Introduction

The CatSper channel is located in the principal piece of the sperm tail and is a unique cation channel whose activity is necessary for sperm hyperactivation and motility [1]. Four *Catsper* (cation channel of sperm) genes encoding different pore-forming α units have been cloned and characterized as essential genes for hyperactivation and male fertility [2,3]. These genes display exclusive expression in male germ cells and form a molecular structure similar to voltage-dependent K^+^ channels; however, CatSper is a calcium-permeable channel [1,4]. In addition, the activity of CatSper channels also relies on the presence of accessory subunits (β,δ, ε, and ζ Catsper), which interact with EFCAB9, a pH sensor protein that triggers sperm hyperactivation [5,6,7,8]. During the sperm capacitation process, CatSper activity increases due to intracellular alkalization. Likewise, CatSper activation produces an intraflagellar Ca^2+^ influx leading to sperm hyperactivation [9]. 

The mouse *Catsper1* gene is the best-characterized member of the family regarding its sequence and activity. It shows >50% identity with rat and human DNA sequences and proteins [1]. In addition, the *Catsper* 2, 3, and 4 genes share characteristics with *Catsper1,* such as α pore-forming subunits and have an exclusive expression in the testis [2,3]. The *Catsper1* knock-out abrogates channel formation and severely affects mouse sperm fertilization capability [1]; similarly, patients harboring mutations in the *CATSPER1* gene show infertility [10]. Therefore, *Catsper1* may be considered a prominent target for the design of male contraceptive strategies due to its exclusive expression in sperm cells, preventing possible side effects in other cell types.

The Sox family comprises transcription factors containing the HMG (High Mobility Group) box. Interestingly, *Sox9* expression has been associated with differentiation of the testis [11,12]. Sox proteins contain transactivation domains and nuclear localization signals. For nuclear transport, Sox proteins have binding domains for CaM and importins [13,14]. In addition, the murine *Catsper1* promoter has elements for functional interaction with Sox, Creb, Crem, and ER transcription factors [15]. Indeed, positive Sox regulation in vitro on the *Catsper1* promoter has been suggested [16]. The *Catsper1* promoter has three Sox sites (A, B, and C), and the mutation of the Sox-B site results in the downregulation of basal transcription in vitro. On the contrary, the presence of Sox9 or Sox5 increases the *Catsper1* promoter activity. Both transcription factors have a synergistic effect, revealing a central role for Sox in the *Catsper1* gene transcription [16]. The *Catsper1* promoter has not been evaluated in the testicular environment [17,18,19,20,21,22], where Sox factors could affect its activity as they do in vitro. 

Likewise, several Sox proteins depend on Calmodulin (CaM) to be transported into the nucleus. CaM is a Ca^2+^-binding protein and a ubiquitous regulator of numerous cell processes, including nuclear transport. Calmidazolium (CMZ) (1-[bis(p-chlorophenyl) methyl]-3-[2,4-dichloro-3-(2,4-dichlorobenzyloxy) phenethyl] imidazolinium chloride), is a potent CaM inhibitor. Interestingly, it has been suggested that CMZ may affect the nuclear transport of Sox proteins containing the CaM-binding domain (Sox9 and Sry) in male germ cells [13,14]. Inhibition of Sox nuclear transport could affect the expression of genes during spermatogenesis and eventually affect the fertility of the male gametes.

Here, we investigated the activity of the *Catsper1* promoter and derived mutants in the Sox binding sites by in vivo transfection of the murine testis. We also studied the effects of the inhibition of Sox protein nuclear transport by CaM on the *Catsper1* gene transcription, sperm motility, and fertilization capacity of sperm. This approach could positively impact human reproduction and economically important species through implementing male contraceptive strategies.

We propose one contraceptive strategy at the molecular level to downregulate Catsper1, which is known to be the principal protein of the calcium channel.

## 2. Results

### 2.1. Catsper1 Promoter Activity in Mouse Testis and Germ Cells

Sox5 transcription factor transactivates *Catsper1* promoter in HEK-293 and MSC-1 cells [16]. In this report, we tested *Catsper1* promoter constructs expressing GFP and Luciferase in mouse testes. The *Catsper1* promoter was cloned in the pIRES-hrGFP-1a vector by removing the CMV promoter and substituting it with the *Catsper1* promoter obtained from the construction pGL3 luciferase vector (pCats) previously obtained. Plasmids maintain the IRES region and GFP ORF under the control of the *Catsper1* promoter with either different lengths or mutations in the Sox sites (Figure 1a and Figure 2a). These new GFP constructs were transfected in mice testes and GFP expression was monitored in vivo. After 72 h, seminiferous tubules were obtained for GFP fluorescence detection in intact tissue. Images showed an evident fluorescence in the seminiferous tubules with all pCatsGFP plasmids compared with the pIRES vector (Figure 1a and Figure 2b). However, pCats798GFP seems to have more fluorescence than the other 1200, 599, and 399 bp *Catsper1* promoter constructs in vivo (Figure 1b). 

Likewise, pCats798GFP constructs harboring mutations in Sox A, B, and C sites of the *Catsper1* promoter were also transfected in testes (Figure 2a). Seminiferous tubules showed slight GFP fluorescence compared to the wild type control and were similar to pIREShr-GFP1a with the CMV promoter (Figure 2b). However, the GFP fluorescence observed in seminiferous tubules indicates that the *Catsper1* promoter produces GFP, but its precise localization in the tubules is unknown. 

The GFP protein expressed from the pCatsGFP constructs containing deletions or mutations in the *Catsper1* promoter was immunodetected in the testis sections to identify cell types. A specific antibody for *Renilla* GFP was used to detect protein in transversal sections from testes transfected with the constructs (Figure 3). *Catsper1* promoters (1200 bp and 394 bp) allowed GFP expression in the cytoplasm of the germ cells. On the contrary, GFP immunodetection of 798 bp and 599 bp *Catsper1* promoters was localized in peritubular cells and Leydig cells, outside the seminiferous epithelium. Additionally, Sox5 immunodetection for the same constructs showed protein expression in somatic cells (peritubular cells and Leydig cells) instead of germ cells, except for a low signal for pCats1200GFP (see Appendix A).

The plasmids with mutated Sox A, B, and C sites in the *Catsper1* promoter were transfected into the testis for GFP immunodetection. The transcriptional activity of mutated plasmids was low for SoxB and SoxC sites, while SoxA seemed to have good GFP expression (Figure 4). GFP was located in somatic cells for the three mutants and only a few signals were observed in pCats798GFP-SoxBmut in the cytoplasm of germ cells. In the same way, Sox5 immunodetection showed signal only in somatic cells (See Appendix A). We decided to quantify the in vivo transcriptional activity of *Photinus* luciferase as a reporter gene from previous constructs (pCatS) harboring the same regions and mutations in the *Catsper1* promoter. 

### 2.2. In Vivo Transcriptional Activity of the Catsper1 Promoter

The transcriptional activity of the *Catsper1* promoter and its variants was explored by the direct activity of luciferase after testicular transfection in vivo. Testis protein lysates were assayed for luciferase activity 72 h after the intratesticular injection of the plasmid. The constructs showed transcriptional activity in the testis; however, the transcriptional activity of the pCatS798 construct was lower than previously observed in vitro (Figure 5a) [16]. The pCatS364 construct, which lacks the Sox-C site of the *Catsper1* promoter (−492 to −288 bp region), showed the highest transcriptional activity in vivo. Instead, the pCatsS 261, 599, and 1200 constructs showed less transcriptional activity in vivo (Figure 5a).

### 2.3. Sox Site Mutations Affect In Vivo Transcriptional Activity

The luciferase-expressing pCatS798 plasmids and their derived mutants at the Sox sites were transfected into murine testes to determine their transcriptional activity. The in vivo transcriptional activities of mutations in Sox-A and Sox-C reduced the transcriptional activity by 50% and 70%, respectively, compared to the wild-type promoter pCatS798 (Figure 5b). Likewise, Sox-B and C mutants showed decreased activity as observed with the GFP fluorescence emitted in transfected seminiferous tubules and GFP immunodetection (Figure 2 and Figure 4). Indeed, the Sox-B mutation decreased the transcriptional activity of the *Catsper1* promoter in vivo by up to 90%.

### 2.4. Effect of CMZ on Catsper Gene Expression

CMZ inhibits the nuclear import of Sox transcriptional factors; therefore, we investigated the in vivo expression of the *Catsper1* gene after CMZ treatment by performing PCR assays as described in the materials and methods. Our results showed that *Catsper1* expression decreased after CMZ exposure (1 μM; 72 h) (Figure 6a). Likewise, the CATSPER1 immunofluorescence of testicular tissue showed reduced Catsper1 protein expression after treatment with CMZ (1 μM) for 96 h (Figure 6b). Treatment with different concentrations of CMZ injected in mouse testes showed a reduction in the *Catsper1* expression with 1 and 7.5 μM CMZ after 72 h (Figure 6c), while lower concentrations still allowed *Catsper1* gene expression. The *Catsper* 2, 3, and 4 mRNA remained stable in the presence of low doses of CMZ. On the contrary, an increase in *Catsper*3 and 4 was observed at the 1 and 7.5 μM concentrations (Figure 6c). These changes in *Catsper* expression might affect channel formation and sperm motility.

### 2.5. Sperm Motility Is Affected by CMZ Treatment

Mice were treated via intratesticular injection with CMZ (1 μM) once a week for five weeks (a whole spermatogenesis cycle) to investigate the effects of the drug on sperm motility. Sperm was obtained from the epididymis and kept in supplemented DMEM-F12 medium to examine the motility of sperm. Motility was first analyzed in the first 72 h of treatment, and no significant changes were observed (Table 1). However, after 1 to 5 weeks of CMZ treatment, the progressive cell motility appeared to be significantly reduced in the first week, with a partial recovery during weeks 2 and 3, but with an increase in the number of cells exhibiting non-progressive and no motility patterns (Table 1). A decrease in progressive motility was evident after four weeks of treatment, and a complete loss of progressive motility was observed in the samples after five weeks of CMZ treatment. In contrast, in the control condition (DMSO-treated samples), the progressive cell motility was preserved after five weeks of treatment (Table 2).

### 2.6. CMZ Effects on Male Fertility

Male mice were treated for five weeks with CMZ or DMSO as a vehicle, and control mice were caged with females to determine their reproductive capacity. Each male mouse was caged with two adult virgin female mice. Each group was monitored for pregnant females, taking 100% of the male fertility rate in the control group mated with female mice. This analysis showed that the number of litters for each male decreased to 30.5% with DMSO (vehicle) and 14.6% after the CMZ treatment (Table 3). The litter/male rate improved slightly (about 16.6%) 8 weeks after finishing the treatment with CMZ (Table 3). In addition, an evaluation of litter size from pregnant females found an average of six pups for each pregnant female mouse in the control group. For the group treated with the vehicle (DMSO), the average was 5.5. Likewise, the litter size of each pregnant mouse in the CMZ-treated group had a reduced number of five pups, which improved to 5.3 eight weeks after finishing CMZ treatment. Taking into account all the tested male mice, a mean for each male renders 6 for each control mouse, 1.83 for each DMSO mouse but only 0.83 for each CMZ treated mouse, and 1.33 for recuperated mice after eight weeks without CMZ. This lack of recovery of fertility leads us to analyze whether there are histological alterations in the testis of treated mice.

### 2.7. CMZ Effects in the Histology of Testis

Testes collected each week after CMZ treatment were fixed and soaked in paraffin for ulterior cut and histological analysis by hematoxylin-eosin staining. Images show transversal seminiferous tubules of the normal testis as the control at 10×, 20×, and 40× magnification (Figure 7a). After the first and second weeks, no visible alterations were observed in the seminiferous epithelium (Figure 7b,c); however, in the third week, we found a disorganized seminiferous epithelium inside tubules (Figure 7d). After four weeks of CMZ treatment, testes showed some tubules with fibrosis, immature germ cells in the lumen of the tubule, vacuolation, empty lumen, intratubular calcification, and thickening of the interstitial cells (Figure 7e). At five weeks under CMZ treatment, testes exhibited evident fibrosis in half the testicular tubules with a borderline with necrosis and intratubular calcification, fading of germ and Sertoli cells. Testes displayed parenchyma invasion into tubules, trapping of sperm, loss of basal lamina in the seminiferous tubules, and fading of interstitial and Leydig cells (Figure 7f). All these alterations in the testis disrupt spermatogenesis in CMZ-treated mice.

## 3. Discussion

Previous in vitro analysis of the *Catsper1* promoter allowed identifying transcriptional regulators and intrinsic elements that mediate transcriptional activation. These observations require validation in vivo using spermatogenic cells, where natural expression of *Catsper1* occurs. The generation of transgenic animals might help to solve this issue; however, the in vivo transient transfection of plasmids could also be helpful to evaluate the *Catsper1* promoter, as has been reported for other promoters of testis-specific genes in mice [21,22]. Deletions of the *Catsper1* promoter region and the effect of the mutations introduced in the three Sox binding sites were evaluated in murine testis. Both the direct observation of GFP in seminiferous tubules and GFP immunodetection in histological testis sections demonstrated the functionality of the *Catsper1* promoter in vivo (Figure 1b and Figure 2b). However, to determine its localization in seminiferous tubules, GFP expression was immunodetected. Only the pCats1215 GFP and pCats364GFP exhibit GFP expression in the cytoplasm of germ cells and behave as germ cell-specific promoters. However, GFP expression from other plasmids was observed also in somatic cells, which indicates that these constructions lack elements for the tissue-specific expression of *Catsper1* (Figure 3). *Catsper1* transcription is dependent on Sox sites of the promoter since mutation of any of three sites decreased transcriptional activity in vivo, implying the participation of Sox family proteins in its regulation (Figure 2 and Figure 4). Several Sox factors, which act as transactivators, contain domains for dimerization and importin or calmodulin-mediated nuclear transport. Sox factors 3, 5, 6, and 17 have been detected in spermatocytes and spermatids in the testis, while Sox factors 3, 4, 12, 13, and 18 locate in spermatogonia. Finally, Sox factors 4, 8, 9, and 12 are present in Sertoli cells of the adult mouse [23]. The inhibition of Sox nuclear transport by CMZ and the decay of *Catsper1* mRNA indicate that these factors could interact with the Sox sites in the *Catsper1* promoter and transactivate its expression in vivo.

Thus, the characterization of the *Catsper1* promoter has allowed us to understand the role of Sox factors in its regulation. In particular, here, we show that the inhibition of Sox nuclear transport prevents *Catsper1* promoter transcriptional activity and gene expression. The decrease in the expression of *Catsper1* precludes the formation of the Catsper ion channel and produces a progressive reduction in motility after five weeks of treatment with CMZ (Table 2) and a reduction in its fertilizing capacity, resulting in fewer litters and pups (Table 3). This infertility effect might be reversed after stopping the CMZ treatment, but only a small recovery of fertility rate was observed (Table 3). This irreversibility in male fertility is due to cellular damage to seminiferous epithelium after 4 and 5 weeks of CMZ treatment, however, some functional tubules still produce sperm (Figure 7). 

Unexpectedly, in the group treated with dimethylsulfoxide-DMSO, a decrease in the offspring to 30.5% was observed, but this should have no effect as it had protective properties. CMZ must be dissolved in DMSO for its administration, however, our results indicate that DMSO may impair male reproductive performance. It has been reported in the literature that DMSO acts as a free-radical scavenger against the effects of radiation and helps preserve some male fertility [24]. Possible causes of impaired male fertility could be the number of injections in testicles and cell damage, since DMSO with DNA used as a transfection agent causes marked histological changes in seminiferous tubules [18].

CMZ can interact with different biomolecules and provoke different effects. For instance, drug administration at micromolar concentrations in osteosarcoma cell cultures has been shown to result in an influx of extracellular calcium and cell proliferation inhibition [25]. In Madin Darby canine kidney (MDCK) cells, CMZ treatment produces extracellular calcium influx and increases the release of calcium from intracellular stores dependent on phospholipases C and A2 activity [26]. Another calmodulin inhibitor, W7, prevents the ATPase activity of the sarcoplasmic reticulum and stimulates the mobilization of intracellular calcium in human umbilical vascular endothelial cells (HUVECs). The drug also produces changes in the expression of the E-Selectin and ICAM-1 gene [27]. Therefore, the effects of CMZ on calcium dynamics after intratesticular administration cannot be ruled out. In addition, the presence of CMZ in vitro affects sperm capacitation, inhibits tyrosine phosphorylation in proteins, and affects sperm motility but not viability [28]. Even though these experiments were performed in mature sperm in vitro, CMZ in the female tract might lead to similar results during capacitation and eventually could be beneficial in contraception. Furthermore, CMZ (>2 µM) stimulates steroidogenesis in primary cultures of rodent Leydig cells. The increase in testosterone and pregnenolone is independent of cAMP, Ca^2+^, and the translation of new proteins. This side effect of CMZ treatment in the testis might also affect spermatogenesis [29]. The increase in the synthesis of steroids by Leydig cells in the testis might affect the expression of *Catsper1* and all four *Catsper* genes.

The inhibition of calmodulin by CMZ prevents the nuclear transport of Sox 9 and SRY, proteins that contain calmodulin-binding domains [13,14]. CMZ might alter the expression of diverse genes under Sox control during spermatogenesis and affect the fertility of male germ cells. Since various Sox proteins (Sox 9, 17, and 30) have well-conserved nuclear localization signals (NLS), the inhibition of calmodulin would prevent their transport to the nucleus and affect *Catsper1* gene transcription among other genes. Indeed, it has been reported that Sox 30 is essential for fertility. Interestingly, the Sox30-null mouse has decreased *Catsper1* expression, despite other transcriptional factors that might regulate its expression [30]. Therefore, the *Catsper1* promoter activity may require calmodulin-dependent nuclear transport of Sox factors. Consistent with this, our data indicate that mutations in the Sox sites in the *Catsper1* promoter prevent its interaction with Sox factors and affect the activity of the *Catsper1* promoter in vivo (Figure 2b and Figure 4).

Inhibition of Sox nuclear transport disrupts *Catsper1* gene expression, and in consequence, the formation of the essential pore-forming α subunit. In this scenario, functional CatSper channels may not be formed, given that other Catsper subunits cannot replace the Catsper1 α subunit, affecting sperm hyperactivation and male fertility [1]. For this reason, Catsper1 has been considered a relevant target for different contraceptive strategies. 

There are very few methods for men regarding male fertility regulation to share equally the burden and benefits of family planning. Among these contraceptive strategies, antibodies directed to the transmembranal domains and the pore region of the Catsper1 channel have been tested in vitro, which produce a significant inhibitory effect on sperm motility and fertility [31,32]. In contrast, a DNA vaccine against Catsper1 provoked a reduced contraceptive effect in male mice [33]. 

Likewise, the transcriptional control of *Catsper1* could be a viable alternative for male contraception. Some transcription modulators developed for the PPAR transcription factors that regulate the transcription of various metabolic genes related to dyslipidemia, and type 2 diabetes mellitus [34], have shown promising results.

Moreover, the intratesticular injection of CMZ limited its effects on the testis, and it prevents side effects in other organs. However, the intratesticular injection could not be an optimal route of administration for humans. Therefore, the study of different administration methods and the use of analogous molecules with specific effects will be necessary to reveal the real contraceptive potential of this strategy.

## 4. Materials and Methods

### 4.1. Animals

For in vivo experiments, Balb/C 8- to 10-week-old mice, sexually mature and suitable for fertilization, were used (*n* = 129 male; 62 female). The animals were maintained in the Animal Facility of the National Medical Center SXXI, IMSS. They were fed a commercial diet (5859, HARLAN) with 9% crude protein (PC) (standard diet) and drinking water ad libitum. Mice received the veterinary care and attention specified in strict accordance with the Mexican Guidelines for Animal Use and Experimentation. The National Scientific Research Committee IMSS approved all experimental procedures and the euthanasia method designed to cause minimal pain and distress, license number R-2014-785-002. Based on the reported in vivo transfection experiments, three animals were the minimum necessary to evaluate each experimental condition and perform the statistical analysis that compares experimental conditions and controls. A maximum of 35 days was the duration of the experiments for male fertility.

### 4.2. Constructs

For in vivo testis transfection in this study, constructs containing the *Catsper1* promoter upstream of the *Photinus* luciferase reporter (pCatS series) were used [16]. A new series of pCats plasmids harboring deletions and mutations of Sox sites in the *Catsper1* promoter was linked to the humanized *Renilla* green fluorescent protein (hrGFP). The pCRTOPO-798Cats was used as the source to substitute CMV promoter by the *Catsper1* promoter region into *Nsi*I-*Bam*HI sites of the pIRES-hrGFP-1a vector to evaluate transcriptional activity in vivo. The resulting construct, pCatS798GFP, allowed observation of the *Catsper1* promoter activity in tissues and was the basic construction for different deletions and mutations.

The plasmid pCatS798GFP was used as a template to generate the Sox site mutations by PCR mutagenesis (Quick-Change site-directed mutagenesis, Stratagene, La Jolla, CA, USA). The plasmid pCats798GFP was the initial template to shorten the region to 599 and 364 bp lengths by PCR mutagenesis, a modification to the directed mutagenesis protocol, which consists of performing the PCR assays in two stages. The first stage includes two separate reactions that contain one of the oligonucleotides (forward or reverse) in a five-cycle amplification. The second stage consists of mixing both reactions to continue PCR amplification for 16 cycles [35]. The fidelity of the cloned *Catsper1* promoter regions and site-directed mutation in the new constructs was confirmed by digestion with specific restriction enzymes and automatic sequencing. All plasmids were transformed and amplified in *E.coli* DH5α, and the plasmid purification was performed with the Maxiprep Plasmid kit (Qiagen, Hilden, Germany).

### 4.3. Catsper1 Promoter Transfection in Adult Mouse Testis

Adult Balb/C mice were anesthetized intramuscularly (IM) with a mixture of xylazine hydrochloride 1 mg/kg and ketamine 10 mg/kg. In addition, preoperative local analgesia was subcutaneously injected into the testis (4 mg/kg of Rimadyl, Pfizer Ltd., New York, NY, USA). For each mouse, the right testis was the experimental organ, while the left testis remained intact and was used as a negative control. The scrotal region was shaved to expose the testis and the epididymis through the skin. A solution containing 25 μg of the experimental plasmid and 3 μL of in vivo TurboFect (Thermo Scientific, Waltham, MA, USA) reagent was mixed in 5% glucose with trypan blue total volume of 20 μL. The transfection mix for each plasmid was injected with an insulin syringe and a 32G ultra-fine needle into the testis. The trypan blue allows visualizing the DNA solution spreading from the injection site to the surface of the testis. For the post-treatment recovery, a single dose of 100 μL of crystalline sodium benzylpenicillin 1,000,000 U/2 mL and 2.5 μL of dipyrone (Dipirone 50-Virbac, 0.5 mL/5 kg) was injected IM in each mouse. 

### 4.4. Catsper1 Promoter Activity by GFP Detection

The plasmid pCatsGFP798 and the three mutated Sox site constructs were transfected into mice testis. The animals were kept under close observation for three days to allow for GFP accumulation and then sacrificed by cervical dislocation to cause a painless death, and the testes were surgically removed. The seminiferous tubules were exposed to a Dulbecco’s Modified Eagle Medium (DMEM) and immediately observed in an inverted Olympus epifluorescence microscope. Subsequently, the images were captured using photographic equipment integrated into the microscope.

### 4.5. Immunofluorescence

The constructs pCatsGFP with deletions or mutations were transfected into the testis following the same procedure. The testes were removed, cryoprotected with isopentane (Sigma Aldrich, Burlington, MA, USA), frozen on dry ice, and stored at −80 °C before immunofluorescence (IF) analysis. All testes were mounted with tissue-tek (Sakura Finetek, Nagano, Japan) in a cryostat microtome GM 1510S (Leica Microsystems, Wetzlar, Germany) and cut into 10 µm sections. Sections were placed onto poly-L-lysine-coated slides and fixed with 4% paraformaldehyde in 0.12 M PBS [pH 7.4] for 24 h, and then washed with 0.12 M PBS and 0.025% Tritón X100 for 5 min. Sections were incubated for 30 min at room temperature in a blocking solution containing 10% bovine serum albumin Fraction V (Roche, Basilea, Suiza) in 0.12 M PBS and subsequently exposed to anti-GFP (Stratagene, San Diego, CA, USA), anti-Catsper1 (H-300), and anti-Sox5 (H-19) antibodies (Santa Cruz Biotechnology, CA, USA), diluted in PBS (1:100) for 24 h at 4 °C. Sections were then rewashed with 0.12 M PBS and 0.1% Tween20 for 10 min, and immediately incubated with anti-rabbit IgG Alexa 486 for GFP and Sox5 (Santa Cruz Biotechnologies, Santa Cruz, CA, USA) antibodies, and anti-rabbit IgG fluorescein for Catsper1 and diluted 1:400 in PBS, for two h at room temperature. Last, the sections were washed with 0.12 M PBS and 0.1% Tween20 for 10 min before adding 100 μL of 300 nM DAPI diluted in 0.12 M PBS for 5 min. Samples were rewashed (0.12 M PBS), dyed, and gently mounted onto coverslips with Vectashield (Vector Labs, Newark, CA, USA) mounting reagent. Images were acquired on a Nikon Ti Eclipse inverted confocal microscope equipped with an A1 imaging system. Imaging was performed using either a 20× (dry, NA 0.8) or 60× (oil immersion, NA 1.4) objective lens. Dyes were excited in a sequential mode using the built-in laser lines: 403 nm (DAPI), 488 nm (Alexa 486) Corresponding fluorescences were read in the following ranges: 425–475 nm (DAPI), 500–550 nm (Alexa 486) with the manufacturer-provided filter sets. Images were acquired and analyzed using software NIS Elements v.5.50.

### 4.6. Analysis of Catsper1 Promoter Activity In Vivo

The different deletions of the *Catsper1* promoter and the *Sox* mutated sites were evaluated by transfecting mice testes with pCatS plasmids containing the Luciferase reporter gene. After three days, the testes were collected and prepared for the luciferase assay. Briefly, the seminiferous tubules were cut into small pieces and suspended in 3 mL of phosphate saline buffer (PBS), mixed by continuous pipetting for 2 min at room temperature to obtain the germ cells. The supernatant was recovered and centrifuged at 2000 rpm to obtain the cell pellet. Cells were suspended in 800 µL of lysis buffer included in the dual luciferase assay kit (Promega, Madison, WI, USA). The luciferase assays were performed using 200 µL of the cell lysate, 200 µL buffer with the substrate (LAR-II) for Photinus luciferase, and 200 µL of STOP-GLO for Renilla luciferase activity were added later. The chemiluminescence activity was measured in a luminometer (Junior LB 95O9, Berthold Technologies, Bad Wildbad, Germany) as relative luminescence units (RLU).

### 4.7. Intra-Testicular Injection of Calmidazolium

The inhibitor of calmodulin, calmidazolium 1 μM (CMZ) was dissolved in 5 mM DMSO as a vehicle and tested by intratesticular injection in male mice. First, a time course was made with 1 μM of CMZ intratesticular injected in mice. Testes were recollected after 24, 48, 72 up to 96 h for total RNA extraction, followed by RT-PCR for *Catsper1* expression and immunofluorescence of Catsper1. Second, a test of different concentrations of 5, 10, 1000, and 7500 nM CMZ in a final volume of 10 µL for each testis was assayed in mice. RNA total was extracted from testes and RT-PCR of four Catsper genes was made. Further, to test male fertility, mice were intratesticular injected once a week with 1 μM CMZ or DMSO 5 mM doses for five weeks. Mice were sacrificed by cervical dislocation to cause minimal pain and distress after each treatment to obtain the testes and epididymis.

### 4.8. CMZ Treatment and Fertility Evaluation

Male fertility was assessed in mice intratesticular injected with 1 μM CMZ per testis or 5mM DMSO as the vehicle. Control groups of untreated mice and DMSO-treated mice were used to evaluate fertilization capability (*n* = 3). Briefly, three separate experiments were performed in groups of six, mice received intratesticular injections of CMZ (1 μM) or DMSO in each testis every week for five weeks. Each mouse received five CMZ intratesticular injections in each testis, spanning the complete spermatogenic cycle. Each male mouse was caged with two adult female mice and the presence of a vaginal plug was observed to verify the mate. Male mice were removed after mating and sacrificed by cervical dislocation performed correctly to cause as painless a death as possible, to obtain the testes and epididymis. Testes were analyzed by standard histological hematoxylin-eosin staining and sperm from the epididymis were evaluated for their motility with microscopy. Eight mice treated with CMZ for five weeks were kept for a second mating after eight weeks without treatment (Recuperates). The mating females were observed to count the number of pregnant females per group, the litter size, and to determine the fertility and reproduction rate in the condition tested. 

### 4.9. RT-PCR Analysis for Catsper Genes

The *Catsper* expression was evaluated by RT-PCR 72 h after CMZ treatment. Testis RNA extraction was performed with 1ml Trizol. RNA was separated with 0.2 mL of chloroform by centrifugation at 12,000× *g* for 10 min. The aqueous phase was recovered, and RNA samples were precipitated with 0.5 mL isopropyl alcohol. The RNA pellet was recovered by centrifugation at 12,000× *g* for 15 min and then washed with 70% ethanol-DEPC water, air-dried at room temperature, and redissolved in water. Final RNA concentration was estimated by spectrophotometry (nanodrop 2000, Thermofisher, Waltham, MA, USA). The first-strand synthesis reaction was performed in a total volume of 20 µL with 2 µg of RNA, random primers, dNTPs mix, and adequate buffer conditions for the multiScribe Reverse Transcriptase (High-capacity cDNA reverse transcription kit, Applied Biosystems, Waltham, MA, USA). Pairs of oligonucleotides yielding a Tm of 60 °C were used to amplify the four *Catsper* genes with products of 90–240 bp: Catsper1F 5′ CTGAGCTAGAGATCCGAGGTG 3′, Catsper1R 5′ AGACTTTGAGAATCCGGAACAGGC 3′, Catsper2F 5′ TTTGTGCTGATGGTTGAAATAGA 3′, Catsper2R 5′ TGAAGCTAAGCAAGATGAACCA 3′, Catsper3F 5′ GCTTTCTTTACTCTCTTCAGTTTGG 3′, Catsper3R 5′ CCCGGCTCACAGTAAACTTC 3′, Catsper4F 5′ AGATCCGAGAGGAACTCAACAT 3′, and Catsper4R 5′ ACATTTTGCTGCTCCTGGTT 3′.

RT-PCR was performed in a 20 µL reaction with 1 µL cDNA, 10 pM of each primer forward and reverse, and Mix GoTaq 2X (Promega, Madison, WI, USA). Conditions for *Catsper* gene amplification by PCR were: 94 °C denaturalization for 1 min, 30 cycles of denaturalization 94 °C 30 s, alignment 60 °C 30 s, extension 72 °C 30 s, and final extension of 72 °C 7 min. A 173 bp product of *β-actin* gene was used as an endogenous control for sample normalization with oligonucleotides β-actinF 5′CACCATGTACCCAGGCATTG 3′ and β-actinR 5′ CCTGCTTGCTGATCCACATC 3′. All PCR products were analyzed by 2% agarose gel electrophoresis.

### 4.10. Sperm Motility Determination

Sperm were obtained from the epididymis and divided into sections in 2 mL of Human Tubal Fluid (HTF) medium supplemented with 10% SFB, progesterone, and calcium and kept in incubation at 37 °C for 20 min. Cell count and viability estimation were carried out in a Neubauer chamber with 10 µL of the sperm suspension and trypan blue to stain non-viable cells. Sperm motility was evaluated under light microscopy, and three types of movement were observed: progressive, non-progressive, and no motility [36]. It is worth mentioning that an indicator of murine sperm hyperactivity is the rapid progressive linear movement, which in the control group can reach 36–40% and, together with the type B movement, approaches 55–50% [36,37]. Motility was evaluated during the first 60 min after obtaining the sample, starting with an aliquot of 10 µL in the Neubauer chamber. Two hundred sperm cells were evaluated for motility in each separate experiment using the 40× objective in different quadrants chosen randomly, and cells were classified in A, B, and C categories, as mentioned above.

### 4.11. Experimental Design

To analyze the transcriptional activity of the *Catsper1* promoter with different deletions or mutations in the Sox sites, each plasmid with GFP or Luciferase reporter genes was intratesticularly injected in mice *(n* = 3). Transcriptional activity was observed by GFP fluorescence in vivo from seminiferous tubules under a fluorescence microscope. On the other hand, similar constructs of the *Catsper1* promoter coupled to the *Photinus* luciferase gene luciferase, activities were measured in protein extracts from the plasmid-transfected testis. The localization of GFP in testicular tissue was immunodetected with a specific antibody to determine the cell types with the activity of the *Catsper1* promoter.

To determine the effect of the calmodulin inhibitor, 1 µM CMZ dose was analyzed over a time course of 24, 48, 72, and 96 h for the *Catsper1* expression by RT-PCR (*n* = 3 each time) and immunofluorescence of Catsper1 in testis sections were observed. Likewise, sperm from epididymis was evaluated for motility in this time course for up to 72 hrs. Different doses of CMZ were injected intratesticular and 96 h after the expression of four *Catsper* genes was evaluated (*n* = 3). 

For the evaluation of fertility, a control group without treatment (*n* = 3), one with the DMSO vehicle (*n* = 3), and 1 μM CMZ of mice (*n* = 33) treated for 5 weeks underwent intratesticular injections with CMZ 1 μM every week, of which three mice were sacrificed weekly for evaluation of motility and histological analysis of the testis (*n* = 15). Mice from a control group (*n* = 3), a group with a DMSO vehicle (n = 3), and a group of mice (*n* = 18) treated with CMZ 1 μM up to 5 weeks, were crossed with two mature and virgin females in a 1:2 male, female ratio (*n* = 36). The seminal plug in the females was verified before removing the males, and the number of pregnant females and the number of pups given birth by each female were observed.

### 4.12. Statistic Analysis

The differences between mice testes transfected with experimental plasmids were estimated by paired t-student tests after determining normal distribution through the Kolmogorov–Smirnov test. Comparison of sperm motility among control, treated and recuperated groups were analyzed with a one-way ANOVA statistical test. T-student tests were used to compare the mean fertility rate and litter size between the control and treated groups. All statistical tests were two-tailed, and *p* values < 0.05 were considered statistically significant.

## 5. Conclusions

Sox factors regulate *Catsper1* gene activity in vivo through three interaction sites present in its promoter. Sox factors depend on calmodulin for their nuclear transport, thus allowing CMZ treatment to affect their nuclear location and, therefore, the expression of *Catsper1*. As a result, CMZ treatment decreased sperm motility after one cycle of spermatogenesis, and male fertility was impaired, low litter number and size were observed. The absence of the *Catsper1* expression may prevent the proper formation of the CatSper channel, thereby decreasing male motility and fertility. Thus, specific CaM inhibitors, which exert this kind of regulation during spermatogenesis, might act as a reversible male contraceptive, affecting the expression of genes regulated by Sox in germ cells. This work only focuses on *Catsper* genes; however, other specific testis genes may be regulated by Sox factors. Future *Catsper1* research as a non-hormonal male contraceptive target would involve the drug design to inhibit transcriptional factors for an effective and selective gene down-regulation. 

## Figures and Tables

**Figure 1 ijms-23-08070-f001:**
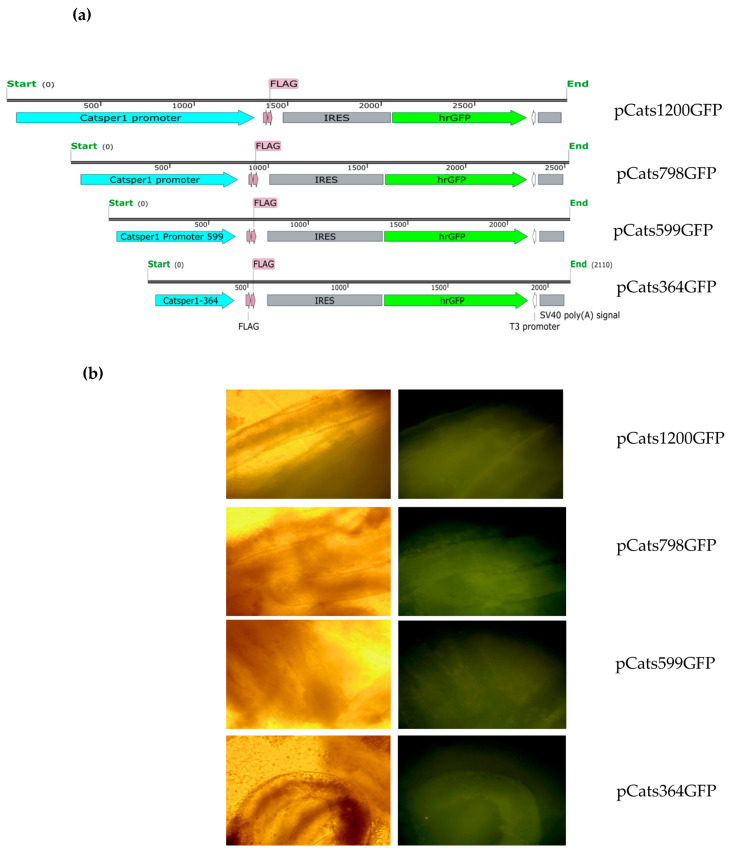
*Catsper1* promoter drives GFP expression in seminiferous tubules. (**a**) The pCatsGFP constructs containing the GFP reporter gene, are shown. The wild-type *Catsper1* promoter and its variants with 1200, 798, 599 and 364 bp length were intratesticular transfected in mice. (**b**) Three days after transfection, seminiferous tubules were isolated by decapsulating the testis and mechanical disintegration. Seminiferous tubules were observed in white light (**left**) and epifluorescence (**right**) microscopy with a 10× magnification objective to show direct GFP fluorescence generated by the Catsper promoter within the tubules.

**Figure 2 ijms-23-08070-f002:**
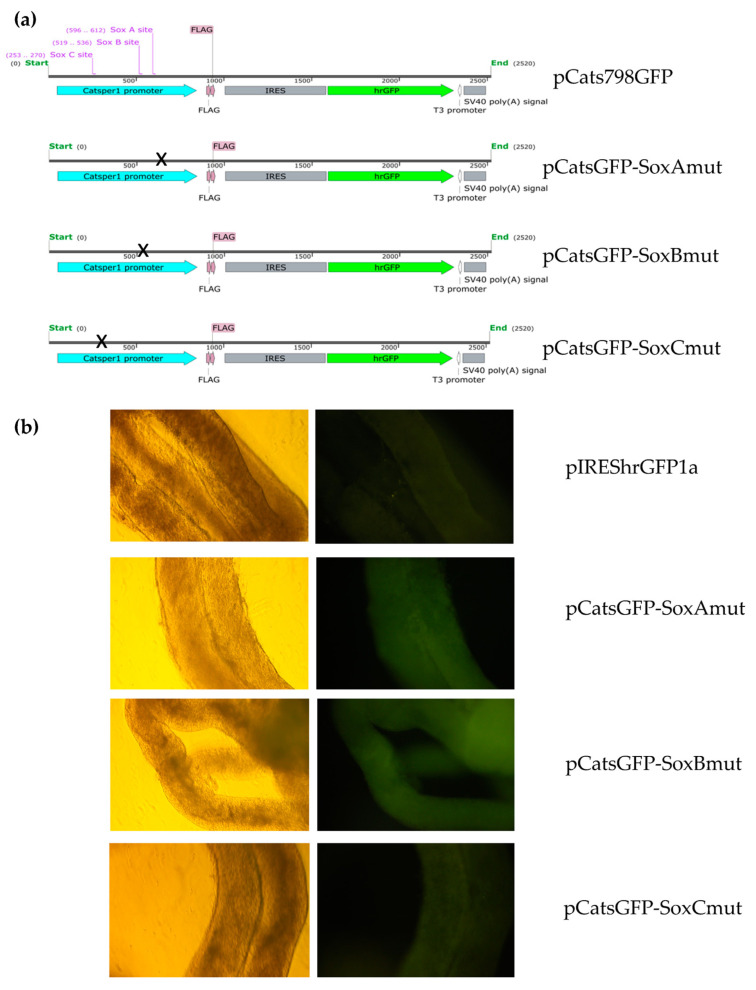
Mutation in Sox sites of *Catsper1* promoter weakens GFP expression in seminiferous tubules. (**a**) The pCats798GFP constructs containing the mutations in the Sox A, B, and C sites maintain the same length, mutations are marked as X in their approximate localization. Each Sox site is indicated above the pCats798GFP map. All constructs contain IRES region (gray) and hrGFP reporter gene (green), *Catsper1* promoter and its mutants are shown in blue. (**b**) After three days of transfection, the testis was isolated, decapsulated, and mechanically separated. Seminiferous tubules were observed in an inverted epifluorescence microscope with a 10× magnification objective. Seminiferous tubules are shown in white light (**left**) and epifluorescence microscopy to show direct GFP fluorescence (**right**) generated by the Catsper promoter with mutations at Sox sites.

**Figure 3 ijms-23-08070-f003:**
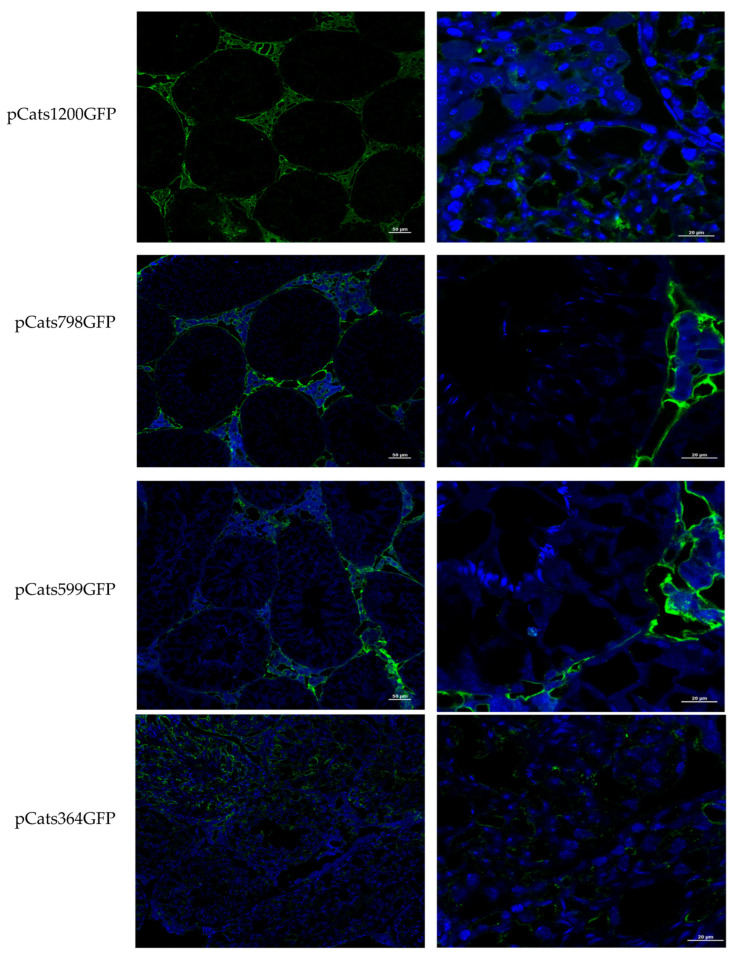
In vivo GFP immunodetection in murine testes transfected with the *Catsper1* promoter constructs. The constructs with different deletions of the *Catsper1* promoter and containing the GFP gene as a reporter gene were transfected by intratesticular injection in mice. The left column shows GFP immunosignals detected with an anti-mouse antibody coupled to Alexa 486 with a 20× magnification. The right column shows images with 60× magnification. Names of the constructs are indicated on the left. The green signal corresponds to GFP and is observed in the cytoplasm of spermatogenic cells inside seminiferous tubules with pCats1200GFP and pCats364GFP. DAPI was used to contrast the cell nuclei.

**Figure 4 ijms-23-08070-f004:**
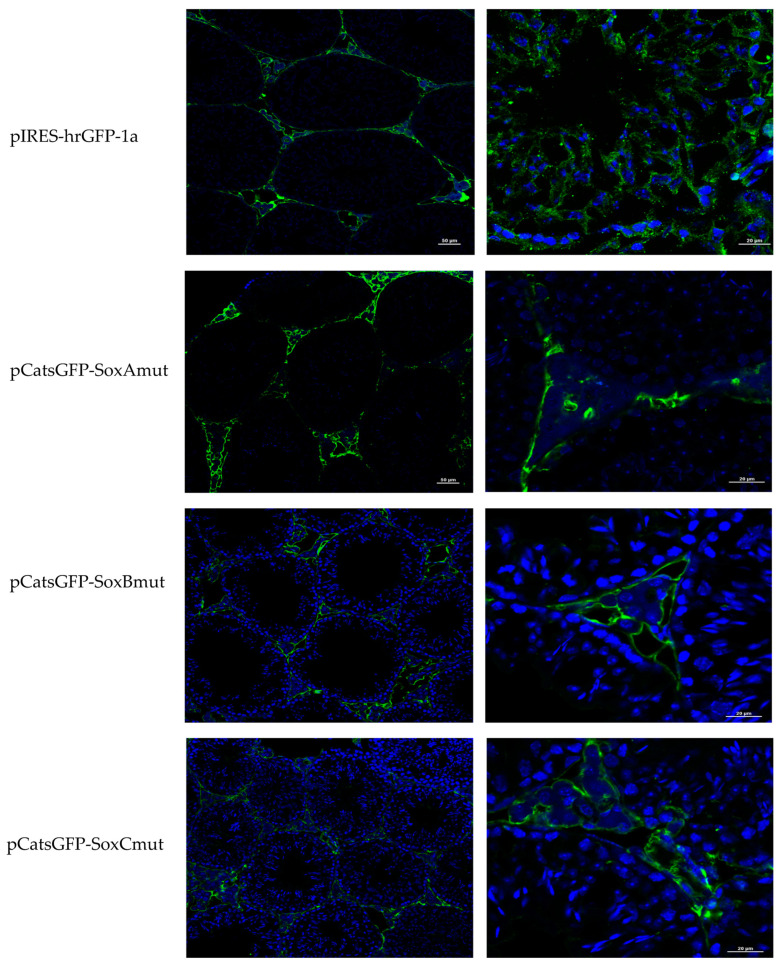
GFP expression in vivo from plasmids with Sox mutations in *Catsper1* promoter. The constructs containing Sox A, B, or C mutations in *Catsper1* promoter and GFP gene as a reporter, were transfected by intratesticular injection in mice. The left column shows 10 µm sections exposed to the Renilla GFP antibody and revealed with an anti-mouse secondary antibody coupled to Alexa 486, preparations were observed in a confocal microscope with a 20× magnification. The right column shows GFP expressed by Sox mutants with a 60× magnification. A green signal is observed in the cytoplasm of somatic cells of the seminiferous tubules, and DAPI was used to contrast the cell nuclei.

**Figure 5 ijms-23-08070-f005:**
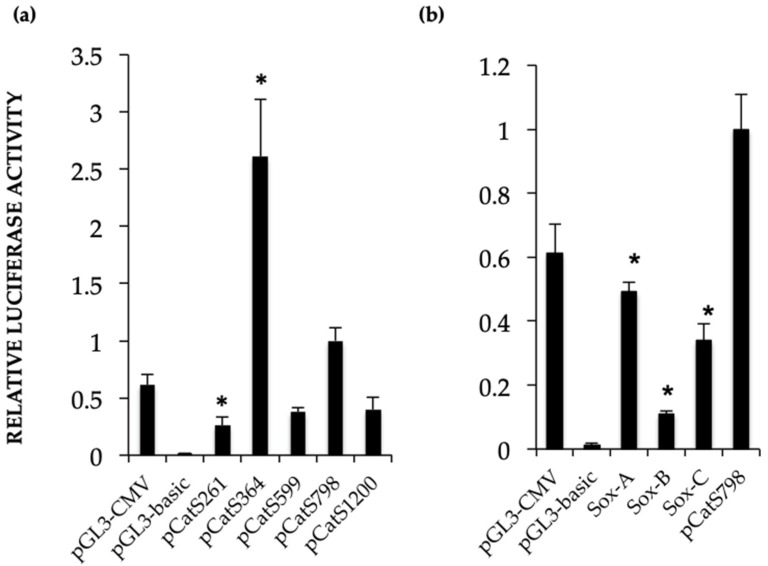
In vivo luciferase activity is induced by the *Castper1* promoter. (**a**) Constructs including the *Catsper1* promoter with deletions that shorten the region are shown from right to left. The 364*-nt Catsper1* promoter resulted in the highest transcriptional activity and pCatS261 the lowest. (**b**) The three Sox mutations introduced in the *Catsper1* promoter have decreased transcriptional activity after transfection in murine testis, in contrast, the mutation in the SoxB site showed the most significant reduction in comparison with pCats798 (*p* = 0.029). Transcriptional activity is expressed in relative units, resulting from the quotient of Luciferase activity of *Photinus* divided by the *Renilla* luciferase activity (in relative units of Light; RUL), normalized to the activity of the promoter pCats798 construct. The results show three experiments for each plasmid performed in triplicate. Bars correspond to the mean ± SEM, and the (*) denotes differences among constructs and the control pCat798 (*p* < 0.05; Student’s *t*-test).

**Figure 6 ijms-23-08070-f006:**
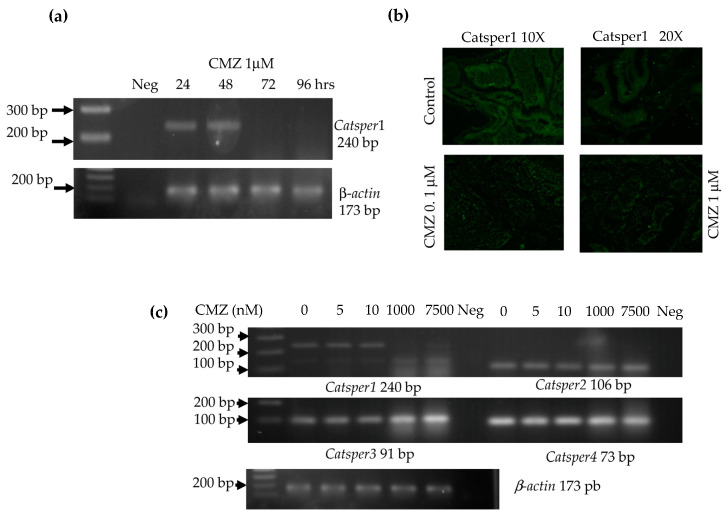
Calmidazolium (CMZ) affects *Catsper1* gene expression. (**a**) RT-PCR of mouse testis treated with CMZ (1 µM) in a follow-up of 96 h after treatment. A 240 bp product was amplified from the *Catsper1* gene by RT-PCR. Negative PCR control with RNA (Neg) (**b**) Immunofluorescence with Catsper1 antibody detected with anti-rabbit IgG fluorescein in fresh tissue slices from wild-type mouse testis before and after CMZ treatment. (**c**) Evaluation of the expression of the four *Catsper* genes in the absence and presence of CMZ, at different doses, as listed, administrated by 72 h. RT-PCR using specific oligonucleotides for each *Catsper* gene and total RNA from wild-type and CMZ-treated mouse testis.

**Figure 7 ijms-23-08070-f007:**
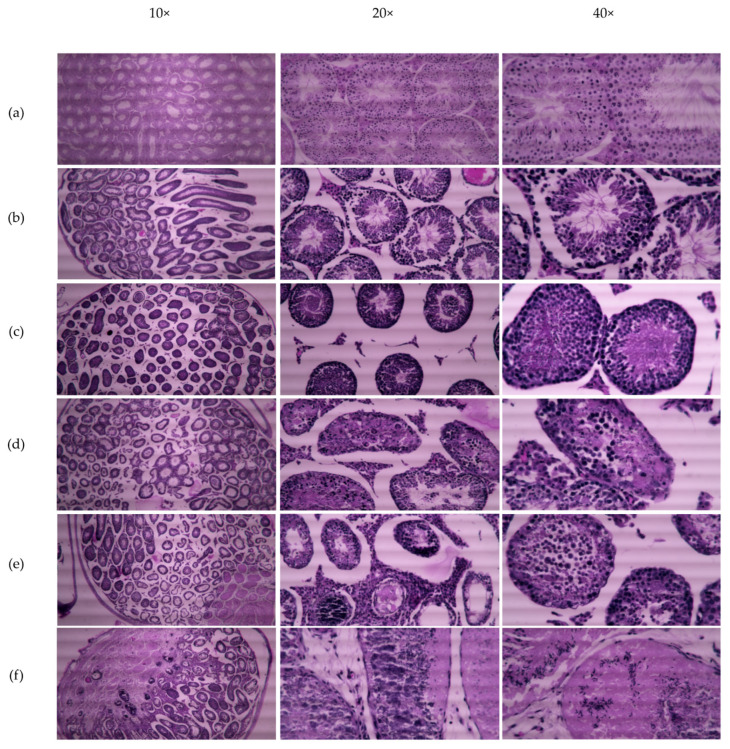
Testicular structure of control and CMZ treated mice. Testis histology of control and CMZ-treated mice after 5 weeks of administration, the treated testes for each week, were stained with hematoxylin-eosin. (**a**) Control testis with normal seminiferous epithelium and interstitial space. (**b**) Testis histology after one week of CMZ without apparent changes in the seminiferous epithelium, lumen, and interstitial cells. (**c**) After two weeks of CMZ administration, the architecture of seminiferous tubules, interstitial space, and germ cells remain without changes. Luminal space is reduced in tubules. (**d**) CMZ treatment for three weeks provokes sloughing or obliteration of lumen by immature germ cells and disorganized seminiferous epithelium. (**e**) After four weeks of CMZ treatment vacuolation, empty lumen, and calcification intratubular are observed, also a thickening of interstitial cells. Germ cell detachment from the basal lamina, exfoliated immature germ cells are in the intraluminal space (**f**) Finally, five weeks with CMZ caused fibrosis in half testicular tubules, with a borderline with necrosis and calcification intratubular, fading of germ and Sertoli cells, parenchyma invasion into tubules, trapping of sperms, loss of lamina basal in the seminiferous tubules, necrosis of interstitial and Leydig cells.

**Table 1 ijms-23-08070-t001:** Evaluation of sperm motility after CMZ treatment (hours).

Motility	Control	24 h	48 h	72 h
Progressive	25.1 ± 2.28	31.21 ± 1.15	15.66 ± 2.61	23.41 ± 4.71
Non-Progressive	36.81 ± 2.14	12.71 ± 1.15	22.89 ± 5.03	21.53 ± 4.8
No motility	37.23 ± 1.96	56.06 ± 0.28	61.44 ± 8.66	55.04 ± 9.94

Values correspond to percentage of analyzed sperm and the mean ± SEM.

**Table 2 ijms-23-08070-t002:** Evaluation of sperm motility after CMZ treatment (weeks).

Weeks	0	1	2	3	4	5	5
**Motility (** **±SEM)**	**Control**	**Calmidazolium Treatment**					**DMSO**
Progressive	25.1 ± 2.28	4.07 ± 0.53	28.64 ± 0.73	28.28 ± 4.05	10.15 ± 0.59	1.10 ± 0.24 ^a,c^	15.77 ± 1.10
Non-Progressive	36.81 ± 2.14	9.65 ± 0.53	27.66 ± 0.24	23.23 ± 3.54	12.94 ± 1.06	39.50 ± 10.72 ^b^	40.06 ± 2.54
No motility	37.23 ± 1.96	86.26 ± 6.88	43.68 ± 0.48	48.48 ± 7.09	76.90 ± 4.0	59.39 ± 4.80 ^a^	44.16 ± 3.92

^a^ Lowest progressive motility (*p* < 0.001; ANOVA); ^b^ Highest non-progressive motility (*p* < 0.05; ANOVA); ^c^ Minor progressive motility than DMSO control (*p* < 0.05; Student *t*-test).

**Table 3 ijms-23-08070-t003:** The fertility rate of CMZ-treated mice.

Group	Males	Pregnants/Females	Pups	Litter Size	Litter/Male	Pups/Male	Mean
Control	3	6/6	36	6	2	12 (100%)	6
DMSO	3	2/6	11 ± 1.16	5.5 ± 0.21	0.66	3.66 (30.5%)	1.83
Calmidazolium	18 ^a^	6/34 *	30 ± 0.38	5 ± 0.06 *	0.3529	1.76 (14.6%)	0.83 **
Recuperates	8	3/16 *	16 ± 0.54	5.3 ± 0.1	0.375	2 (16.6%)	1.33 **

^a^ One male mouse died during mating. ±SEM; * Low fertility rate and litter size than the control group (*p* < 0.001; Paired *t*-test); ** Low mean of pups for pregnant females than the control group (*p* < 0.001; Paired *t*-test).

## Data Availability

Data are available from the corresponding upon request.

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
