# Peer review of "Down Regulation of Catsper1 Expression by Calmodulin Inhibitor (Calmidazolium): Possible Implications for Fertility"

_ijms, 2022, doi:10.3390/ijms23158070_

Round 1
Reviewer 1 Report
In this study, authors investigated the activity of the Catsper1 promoter and derived mutants in the Sox binding sites by in vivo transfection of the murine testis, focusing on the effects of the inhibition of Sox on catsper1 gene transcription, sperm motility, and fertilization capacity of sperm. The study is very articulate, yet it is well described and explained, resulting easy to read and follow even for scientists who are not molecular biologists. The English is excellent. Moreover, this research may represent the base for future male contraceptive strategy at the molecular level. Considering its relevance and originality, I recommend its publication after minor revision. Please see specific comments below.
Kindly remove all extra spaces between the words.
Line 50: there are empty parenthesis.
Lines 102-105: they do not read well after the aim, and do not add nothing specific. I suggest deleting these lines or move to the discussion, or before line 95.
Quality of the figures is poor; they are blurred and unreadable. Please improve the legend of fig 1B: it is unclear what I am looking at.
Lines 167-168: Are all those “????” needed?
Line 171: remove the comma “,” after “promoter”
It would be easy if methods were described before the results, especially in this type of study where different constructs are used. IT would be useful to understand who is who. However, I know it can be a journal request for formatting; in that case, it doesn’t matter.
Fig 6: the legend on top of Western Blotting is not aligned with the lines in fig A and C. Why the controls in fig A and C do not express actin?
Line 345: edit as “behave”.
Lines 372-374: the sentences about WT are not relevant, as it is not used in this study and shift the focus from CMZ.
Discussion is very good, but your results are described after almost 2 pages. I suggest reorganizing the paragraphs to report your results first (lines 416-423), followed by the explanation about the importance in male contraception and all the rest.
The number of litter for each male decreased to 30.5% with DMSO. This should be discussed, and an explanation provided.
Lines 451-452 should not be bold.
Line 614: no capital letter for “progressive”.
In conclusions, summarize your main results.
Reviewer 2 Report
It is really interesting and important study with new insights. Authors proposed a new method based on the application calmidazolium – calmodulin inhibitor. I suggest that the level of study is very high, especially with plasmid implenetation for testis in vivo.
I have only one question – Catsper is calcium-permeable channel, but a lot of mix for transfection contains extremely high calcium concentration. How did authors exclude its influence for observed results? I think that authors should discuss this point.
Author Response
Reviewer 2
It is really interesting and important study with new insights. Authors proposed a new method based on the application calmidazolium – calmodulin inhibitor. I suggest that the level of study is very high, especially with plasmid implenetation for testis in vivo.
I have only one question – Catsper is calcium-permeable channel, but a lot of mix for transfection contains extremely high calcium concentration. How did authors exclude its influence for observed results? I think that authors should discuss this point.
Answer: The authors appreciate the reviewer's opinion on our work, and we greatly appreciate his/her comments. The reviewer's question is interesting and becomes relevant to in vivo transfection experiments of testes. Indeed Catsper behaves as a calcium-permeable channel; however, it is more dependent on intracellular pH for its activation in mature sperm. The intracellular pH change that activates Catsper occurs during capacitation. In our study carried out in testes with immature germ cells, Catsper activation does not occur even in the presence of a high calcium concentration introduced by the transfection mixture. In addition, our studies show the transcriptional expression of GFP directed by the promoter of the Catsper1 gene and not the activity of the protein, so there would be no interference with the results shown. Immunodetection of the Catsper channel was performed on CMZ-treated tissue, which was only incubated in DMSO and PBS buffer, where there is no additional calcium.
